# Effects of Obesity and Calorie Restriction on Cancer Development

**DOI:** 10.3390/ijms24119601

**Published:** 2023-05-31

**Authors:** Ekaterina Sergeeva, Tatiana Ruksha, Yulia Fefelova

**Affiliations:** Department of Pathological Physiology, Krasnoyarsk State Medical University, No. 1 P. Zheleznyaka Str., 660022 Krasnoyarsk, Russia or tatyana_ruksha@mail.ru (T.R.); fefelovaja@mail.ru (Y.F.)

**Keywords:** calorie restriction, obesity, cancer

## Abstract

The risk of malignant tumor development is increasing in the world. Obesity is an established risk factor for various malignancies. There are many metabolic alterations associated with obesity which promote cancerogenesis. Excessive body weight leads to increased levels of estrogens, chronic inflammation and hypoxia, which can play an important role in the development of malignancies. It is proved that calorie restriction can improve the state of patients with various diseases. Decreased calorie uptake influences lipid, carbohydrate and protein metabolism, hormone levels and cell processes. Many investigations have been devoted to the effects of calorie restriction on cancer development in vitro and in vivo. It was revealed that fasting can regulate the activity of the signal cascades including AMP-activated protein kinase (AMPK), mitogen-activated protein kinase (MAPK), p53, mTOR, insulin/ insulin-like growth factor 1 (IGF1) and JAK-STAT. Up- or down-regulation of the pathways results in the decrease of cancer cell proliferation, migration and survival and the increase of apoptosis and effects of chemotherapy. The aim of this review is to discuss the connection between obesity and cancer development and the mechanisms of calorie restriction influence on cancerogenesis that stress the importance of further research of calorie restriction effects for the inclusion of this approach in clinical practice.

## 1. Introduction

Oncological diseases represent one of the leading causes of mortality in the world. Metabolic alterations in cancer cells, in cells within the tumor microenvironment and in the organism in general are crucial conditions of cancer progression. Obesity is one of the most important factors affecting cancer development and progression. It has been proven that excess body weight is associated with an enhanced risk of cancer development in at least 13 localizations, including adenocarcinomas of the endometrium, esophagus, kidney and pancreas; hepatocellular carcinoma; gastric cardia cancer; meningioma; multiple myeloma; colorectal and postmenopausal breast cancers and cancers of the ovarium, gallbladder and thyroid gland [1]. The influence of obesity on melanoma progression is still unclear. There are some investigations that show no correlation between body mass and the risk of melanomas [2,3,4]. On the other hand, obesity is associated with various alterations in metabolism—in particular, lipid metabolism. It was found that the increased transfer of fatty acids from adipocytes to melanoma cells enhanced melanoma metabolism through fatty acid oxidation, which led to the promotion of melanoma aggressiveness [5]. A reduced calorie intake is associated with a prolonged lifespan and a decreased risk of age-related diseases and cancer [6]. In this review, we discuss the metabolic alterations connected with cancer and obesity and the pathways underlying an altered metabolism as the mechanisms of calorie restriction influence.

## 2. Metabolic Alterations Caused by Obesity Leading to Cancer Development

It is suspected that obesity can stimulate various mechanisms involved in cancer initiation and progression. These mechanisms are associated with alterations in carbohydrate and lipid metabolism, abnormalities of the IGF axis, alterations in hormone levels, chronic inflammation and hypoxia.

### 2.1. Carbohydrate Metabolism and Warburg Effect

High levels of glucose are often revealed in individuals with an excessive body mass [7,8,9]. This can be crucial for cancer development due to the Warburg effect. In tumor cells, an increase in glucose uptake and an increased level of lactate formation is revealed. These are caused by the Warburg effect, one of the most important peculiarities of cancer cell metabolism. In normal cells, glucose is transformed to pyruvate, which is used in the TCA cycle for oxidative phosphorylation in the presence of oxygen and is followed by minimal lactate production [10]. The biological significance of the Warburg effect is connected with the enhanced anabolism and increased proliferative activity of cancer cells. The increased production of multiple glycolytic intermediates proposes the use of carbon sources for the synthesis of nucleotides, fatty acids and amino acids in cancer cells [11]. The oxidation of glucose-6-phosphate in the pentose phosphate pathway (PPP) provides ribose 5-phosphate for nucleotide and erythrose 4-phosphate for amino acid biosynthesis. Glucose-6-phosphate, which undergoes oxidation in the PPP, also leads to the generation of NADPH. Further, NADPH can be used for lipid biosynthesis. Glucose metabolism is not the only type of metabolism that can be connected with processes important for cancer development. Other carbohydrates, for example, fructose, are involved in similar mechanisms. Thus, the overexpression of the fructose transporter GLUT5 and enhanced fructose utilization are associated with cancer progression [12]. It was revealed that the fructose transporters GLUT5 and GLUT9 are up-regulated in prostate cancer cells. The increase in fructose levels stimulated pathways associated with cancer cell proliferation and invasion. These pathways include TGFβ2, WNT4 and FGF5 [13]. The increased expression of aldolase B in colorectal cancer cells enhanced the fructose metabolism, which resulted in the promotion of metastasis [14]. Glucose can be metabolized by virtually every cell type in the human body; fructose is mainly metabolized in the liver. The effectivity and the direction of a metabolic reaction depends on liver enzyme activity, metabolic levels of various cofactor substrates and products. GLUT5, a fructose specific transporter, is localized mainly in the small intestine [12]. It can be supposed that liver and small intestine disfunctions connected with obesity may contribute to carbohydrate metabolism alterations.

Thus, obesity impairs carbohydrate metabolism, and high levels of glucose and fructose are associated with cancer initiation and progression via the modulation of pathways regulating cancer cell proliferation, survival and metastasis.

### 2.2. Lipid Metabolism

Alterations to lipid metabolism can be used to support cancer cell proliferation, survival, migration and invasion [15]. In many types of cancer, including breast, ovarian, gastric and prostate cancers, a high expression of CD36 (fatty acid translocase) was found [16]. CD36 is one of the known fatty acid protein transporters in the plasma membrane. It has been revealed that diets with a high amount of fat can induce the expression of NF-κB-dependent CD36 and elicit the O-GlcNAcylation of CD36 at S468 and T470. CD36 is a fatty acid receptor that has been identified as an initiator of metastasis in many types of malignancies. A high CD36 expression drives gastric cancer cell metastasis [17]. An enhanced expression of LDL receptors (LDLR) is associated with a poor prognosis in cases of a malignant tumor in the lung. High levels of LDLR expression were associated with the development of breast cancer. An increase in the LDLR expression was revealed in glioblastomas [18,19]. It was shown that high extracellular lipid levels and LDL cholesterol availability, which resulted in an increased uptake of extracellular cholesterol, was required to support prostate cancer cell proliferation [20]. Increased levels of LDL-c are positively associated with cancer progression in patients with breast cancer or colorectal cancer [21,22]. High cholesterol levels were revealed in patients with melanomas [23]. In contrast to LDL levels, low HDL levels are associated with an enhanced risk of cancer [24]. Cancer-associated adipocytes can enhance lipolysis in response to cancer cell signals, which leads to the increased release of fatty acids. Cancer cells take up these acids, leading to enhanced fatty acid oxidation uncoupled from ATP production that triggers the activation of AMPK signaling [25]. Furthermore, it was shown that high levels of fatty acids in circulation resulted in the increased proliferation and aggressiveness of cancer cells, which was associated with the up-regulation of the estrogen receptor (ER) and mTOR signaling cascades [26].

### 2.3. Hyperinsulinemia, Insulin Resistance and Abnormalities of IGF Axis

It is known that an increased body mass is associated with increased levels of insulin and insulin resistance. Besides this, alterations to IGF and insulin-like growth-factor-binding protein (IGFBP) expression levels were revealed in patients with obesity [27]. The abnormal expression of IGFs, increased insulin levels and insulin resistance trigger the mechanisms associated with the development of various malignant tumors. Chau et al. have showed that patients with an increased body mass index (BMI) had higher IGF levels and a higher risk of prostate cancer development. This can be connected with the mitogenic and antiapoptotic effects of IGFs. It was shown that IGFBP-like protein 1 (IGFBPL1), a member of the IGFBP family, suppressed the clonal formation and proliferation of esophageal cancer cells and triggered cell apoptosis and G1/S-phase arrest [28]. Increased expression levels of IGF2, IGF1 and IGFR1 were revealed in animal models of hepatocellular carcinoma (HCC). It was assumed that they promoted hepatocyte proliferation by means of a paracrine mechanism in the precancerous stage and cancer cell proliferation through an autocrine mechanism [29]. In in vitro investigations, it was shown that IGF1 stimulates DNA synthesis, cyclin D1 expression and the inhibition of proteasome-mediated cathepsin B (CTSB) degradation, which leads to HCC progression and metastasis. IGF2 also takes part in hepatocancerogenesis to enhance neoangiogenesis. The overexpression of IGF2 is associated with HCC hypervascularization. In hepatomas, the increased expression of insulin-like growth factor receptor (IGFR) is correlated with the stimulation of cancer cell proliferation, survival and migration, and the decreased expression of IGFBP1 enhances the hepatoma cell invasion process. It was revealed that the signaling pathways involved in the regulation of IGF effects on the development of hepatomas include the PI3K/AKT and JAK-STAT pathways [30]. It was shown that alterations in the IGFBP expression levels are correlated with gastric cancer prognoses, and the influence on gastric cancer development is connected to the extracellular matrix. The increased expression of IGFBPs is supposed to lead to alterations in focal adhesion, extracellular matrix structural constituents, the extracellular structure and matrix organization [31]. Higher levels of IGFBP4 were revealed in patients with lung cancer compared to a normal, healthy control group. It is known that IGFBPs bind to IGFs and regulate IGF activity. The epigenetic suppression of IGFBP-4 is suggested to positively affect tumor development by reducing IGF inhibition [32]. On the other hand, an IGF-independent mechanism is associated with the IGFBP-4 regulation of tumor growth by means of the modulation of estrogen receptor α activation [33]. It was shown that an increased level of IGF2 is associated with a higher breast-cancer-specific mortality, whereas a higher level of IGFBP is associated with a lower risk of disease recurrence [34]. A high serum IGF-1/IGFBP-3 ratio leads to an increased risk of colorectal cancer. Both IGF-1 and IGFBP-3 can enhance vascular endothelial growth factor (VEGF) gene transcription, which stimulates angiogenesis [35]. Higher levels of IGF-1 and IGF-2 were found in patients with an adrenocortical adenoma (ACA). The promotion of ACA growth by IGFs is assumed to be connected to their regulation of cell growth, differentiation, proliferation and survival [36]. Hyperinsulinemia triggers cancerogenesis via insulin receptor or IGF mediation. Insulin can decrease IGFBP levels, which results in increased IGF activity. It was revealed that the signaling pathways involved in the realization of procancerogenic effects include the PI3K/AKT, mTOR and MAPK pathways in colon cancer cells [37]. The phosphoinositide 3-kinase PI3K/AKT pathway is one of the most important pathways activated under cancer development. The downstream activators of this signal cascade are receptor tyrosine kinases (RTKs), cytokine receptors, integrins and G-protein-coupled receptors (GPCRs) [38]. Metabolic alterations associated with the dysregulation of this pathway result in the promotion of cell survival, growth and proliferation. The PI3K/AKT pathway can influence metabolic processes directly through the phosphorylation-mediated regulation of metabolic enzymes or indirectly through the control of various transcription factors. The main consequences of AKT cascade activation include enhanced glucose uptake and glycolysis, which, in turn, provide metabolic precursors for nucleotide and lipid synthesis. In addition, metabolically activated glucose, in the form of glucose-6-phosphate, enters the hexosamine pathway and pentose phosphate pathway, which leads to the increased production of NADPH used in lipid synthesis and in the regulation of the redox balance [39]. It was revealed that the IGF–AKT pathway plays a crucial role in melanoma development by regulating autophagy [40]. Furthermore, IGF signaling is associated with BRAF inhibitor resistance [41]. Thus, there are several signaling pathways which are supposed to influence cancer development (Figure 1).

### 2.4. Alterations to Sex Hormone Levels

Disturbances in sex hormones also play an important role in tumor development and progression. It is known that obesity is characterized by increased aromatase activity, connected with breast adipocyte hypertrophy, which leads to higher levels of estrogen [42]. An increased body mass and high levels of estrogen are associated with an increased risk of breast cancer in postmenopausal women, but it was revealed that excess adiposity in women with a normal body mass also leads to an elevated risk of invasive breast cancer development in the postmenopausal period. The majority of the patients were estrogen-receptor positive [43,44]. The inhibition of ERα and the stimulation of progesterone receptor (PGR) expression levels in diabetic and non-diabetic obese women with endometrial cancer resulted in an overall improved and progression-free survival rate [45]. Estrogen enhances IGF1 production in the endometrium, which leads to the stimulation of cell proliferation and the inhibition of apoptosis, whereas progesterone increases IGFBP1 synthesis, which results in IGF1 inhibition [46,47]. A high level of testosterone in postmenopausal women is an established risk factor for breast cancer. It was revealed that endurance training leads to a significant decrease in free testosterone [48]. The role of androgens in prostate cancer pathogenesis is still unclear. Testosterone-suppressive treatment can improve survival rates in patients with castration-resistant prostate cancer [49]. On the other hand, there are investigations demonstrating that low levels of testosterone are associated with more aggressive prostate cancer [50,51]. It is known that obesity reduces the testosterone level. An obesity-connected testosterone level decrease is accompanied by lower levels of luteinizing hormone (LH), whereas hypotestosteronism is associated with age and is followed by an increased LH level. This stresses the crucial role of central regulatory mechanisms at the neuroendocrine level in obesity-associated testosterone suppression [52]. In women, higher levels of androgens are often associated with obesity [53]. It was revealed that increased levels of androgens are strong risk factors for endometrial cancer [54]. Moreover, low concentrations of androgens and high concentrations of estrogens are supposed to promote inflammation [55].

### 2.5. Chronic Inflammation

Chronic low-grade inflammation and oxidative stress connected with obesity also take part in tumor development and progression [56]. Under obesity, adipocytes acquire a pro-inflammatory phenotype [57]. Inflammation leads to the enhanced production of mutagenic agents, including reactive oxygen species (ROS), which promote cancer development [58]. Obesity promotes adipocyte dysfunction and cell death [57]. Dysfunctional adipocytes and tissue-resident immune cells enhance the production of inflammatory adipokines and cytokines, including IL-1b, IL-2, IL-6, Il-12 and TNF alpha. IL-1b is supposed to be a communication factor between osteoblast and osteoclast cancer cells. It stimulates the cancer cell transition from a dormant state into an active state and metastasis formation [59]. It has been revealed that IL-6 suppresses cell proliferation via the stimulation of JAK–STAT signaling, but, on the other hand, this cytokine, together with the epidermal growth factor receptor (EGFR), can synergistically influence the up-regulation of the MAPK and PI3K/AKT pathways, resulting in enhanced cancer cell migration [60]. The IL-6-induced stimulation of angiogenesis and inhibition of apoptosis are associated with the modulation of the JAK2–STAT3, PI3K/AKT and ERK signaling cascades [61]. TNF alpha regulates the VEGF activity via the modulation of NF-κB and the ERK1/2 pathway, which results in the stimulation of angiogenesis. Thus, under obesity, adipose tissue and adipose tissue-associated immune cells produces inflammatory factors, promoting tumor progression.

### 2.6. Adiponectin and Leptin Levels

Many factors associated with obesity have influences on the metabolic pathways in cancer cells and the cells of the tumor microenvironment. The consequences of adipocyte dysfunction include a decrease in adiponectin production. Adiponectin is adipokine produced by adipocytes and it is involved in the regulation of many metabolic processes, including insulin signaling, glucose intake and fatty acid oxidation. It is known that adiponectin presents anti-inflammatory and insulin-sensitizing properties. This means that this adipokine can both prevent inflammation and promote its development [62]. An increased adipose tissue mass is positively correlated with the leptin level. Leptin stimulates the secretion of multiple pro-inflammatory cytokines such as IL-1, Il-6, Il-12 and TNF alpha. In turn, some pro-inflammatory cytokines can trigger leptin production in adipocytes by the induction of leptin gene expression, which supports chronic low-grade inflammation [63]. It was revealed that leptin not only enhances melanoma cell proliferation, but also decreases dacarbazine treatment effectiveness by means of the up-regulation of fatty acid synthase (FASN) and heat shock protein 90 (Hsp90). FASN is involved in the regulation of dacarbazine (DTIC) activity and Hsp90 is associated with drug-resistant phenotypes [64]. In breast cancer cells, leptin and estradiol down-regulate AMPK signaling, but insulin, leptin and estradiol activate the PI3K/AKT signaling cascade, which promotes cancerogenesis [65]. Other adipokines can also modulate tumor progression. The overexpression of visfatin or nicotinamide phosphoribosyltransferase (NAMPT) was revealed in the tumor tissues of patients with breast, pancreas or kidney cancers; they are associated with enhanced cancer cell growth, migration and invasion and the stimulation of angiogenesis [66]. These effects were connected with the enzymatic activity of NAMPT or the up-regulation of the PI3K/Akt, MAPK and STAT3 signaling pathways [66,67]. An increased resistin level is associated with the promotion of cancer cell survival and proliferation via the activation of PI3K/Akt and MAPK. The pro-inflammatory effects of resistin, along with the up-regulation of TLR4 and NF κB and an increased level of IL-6, paracrinically induce epithelial-mesenchymal transition (EMT) and metastasis [68]. Apelin also promotes cancer progression via enhanced cancer cell proliferation, migration, invasion and angiogenesis [69]. Hence, disbalance of adipokines produced by disfunctional adipose tissue plays one of the key roles in cancer development.

### 2.7. Hypoxia

An excessive body mass leads to the expansion of adipose tissue, which results in hypoxia and remodeling-induced senescence [70]. These conditions are also associated with the development of endoplasmic reticulum (ER) stress and increased ROS production in adipocytes. ER stress is associated with the impairment of ER-connected processes, including protein folding. Severe ER stress can lead to apoptosis inhibition and autophagy induction [71]. The stabilization of hypoxia-inducible factor 1α (HIF1α) by hypoxia promotes aerobic glycolysis and increases glucose uptake, which, in turn, stimulates protein and nucleotide synthesis. The consequences of these processes are increased breast cancer cell proliferation and survival. Moreover, hypoxia leads to the stabilization of p53, which results in the dysregulation of p53 target genes [70].

Thus, obesity is associated with various factors that initiate metabolic alterations in cancer cells and tumor microenvironment cells by the up- or down-regulation of various signaling pathways, which results in cancerogenesis (Table 1).

Therefore, a decrease in the influence of adiposity risk factors and strategies to combat excessive body weight are very important in preventing malignant tumor development.

## 3. Mechanisms Associated with Calorie Restriction Influence

The important role of obesity in cancer development makes it necessary to search for effective weight loss strategies.

### 3.1. Alterations to Lipid, Carbohydrate and Protein Metabolism

Calorie restriction decreases insulin and glucose levels and improves insulin sensitivity. In addition, gluconeogenesis is enhanced, and glucose oxidation is decreased at the expense of increased fat and protein oxidation [72]. Calorie restriction resulted in a significant decrease in the total cholesterol, triglyceride and LDL cholesterol levels and the ratio of total cholesterol to HDL cholesterol. The level of HDL cholesterol was increased [73]. An improvement in lipid metabolism under the influence of calorie restriction was revealed by Park et al.; it was shown that fasting decreased the liver triacylglycerol levels and serum fetuin-A levels [74]. Calorie restriction in healthy males led to an increase in the erythrocyte omega-3 and omega-6 polyunsaturated fatty acid content and a decrease in the oxidative stress, assessed by the levels of a marker of lipoperoxidative damage, malondialdehyde (MDA). The levels of transferrin also decreased, but the levels of vitamin D increased [75].

Untargeted analyses of the plasma metabolome and the peripheral blood mononuclear cell proteome of healthy, non-obese individuals exposed to long-term day fasting showed a statistically significant increase in polyunsaturated free fatty acids, alpha-tocopherol and a type of vitamin E. The decreased metabolites consisted mainly of amino acids or related metabolites, including ornithine, citrulline and taurine. The level of the pro-aging amino-acid methionine was also depleted. Fasting led to an increase in the beta-hydroxybutyrate levels. It is suspected that increased beta-hydroxybutyrate levels are beneficial for various health parameters [76].

### 3.2. MicroRNAs

Calorie restriction can influence post-transcriptional regulation. A calorie-restricted diet altered the subcutaneous adipose tissue (SAT) microRNA (miRNA) profile. miR-210, miR-132, miR-29a, miR-34a, miR-132 and miR-378 were down-regulated [77]. A decreased expression of microRNA is associated with alterations to enzyme activity and, in turn, the reprogramming of metabolism. Thus, in obese patients, an increased expression of miR-29a-3p after calorie restriction led to lipoprotein lipase (LPL) reduction; a decrease in miR-454-3p led to the up-regulation of acyl-CoA synthetase long-chain family members 1 and 4 (ACSL1 and 4) as well as STAT3; the down-regulation of miR-20b-5p led to the up-regulation of monoglyceride lipase (MGLL), soluble carrier family 2 member 4 (SLC2A4) and STAT3; and a decrease in miR-210 led to the up-regulation of glycerol-3-phosphate dehydrogenase 1-like (GPD1L) [78].

### 3.3. Oxidative Stress and Inflammation

Oxidative stress is a crucial factor in the development of diseases such as cardiovascular and neurodegenerative conditions, diabetes and cancer. Aging is also associated with oxidative stress. As was previously mentioned, calorie restriction can play a role in the regulation of inflammation and oxidative stress. It was revealed that calorie restriction leads to an improvement of biomarkers associated with inflammation (leptin, adiponectin, FABP4, NGAL, OPN and PTX-3) in individuals with type 2 diabetes [79]. Moreover, long-term day fasting in healthy, non-obese individuals resulted in lower levels of sICAM-1, a biomarker associated with various age-associated diseases and inflammation [76]. Under an adipose tissue transcriptome analysis during calorie-restriction-induced weight loss, changes in the BMI were negatively correlated with changes in the mRNA level of genes connected with inflammatory responses, including LIPA, CD68 and GDF15. It was revealed that these genes are specific to anti-inflammatory macrophages [80]. Calorie restriction reduced the oxidative stress in healthy, non-obese, men and women, which was assessed using the urinary concentrations of F2-isoprostanes and validated oxidative stress markers [81]. Calorie restriction decreased the F_2_-isoprostane and IL-6 concentrations in patients with chronic kidney disease. It is known that F_2_-isoprostanes are new biomarkers for oxidative stress. Their generation is associated with the free-radical-catalyzed peroxidation of membrane-bound arachidonic acid [82]. Calorie restriction resulted in a decrease in the tumor necrosis factor-α and interleukin-8 levels in overweight/obese young adults, whereas these markers showed no significant difference in a subgroup of participants with a normal body weight [83]. A protein- and calorie-restricted diet up-regulated the NRF2-mediated stress response. This signaling pathway is responsible for antioxidative and antitoxic defenses. Transcriptional factor NRF2 regulates the expression of the genes involved in oxidative stress defense and xenobiotic detoxication [84]. Calorie restriction increased the CYP2C19 activity. CYP2C19 is an enzyme protein and a member of the CYP2C19 subfamily of the cytochrome P-450 superfamily. CYP2C19 plays an important role in drug metabolism and xenobiotic detoxication [85]. A calorie-restricted diet resulted in the stimulation of P-450 CYP1A2 in obese postmenopausal women [86].

### 3.4. Adipokines and Hormones

Calorie restriction is supposed to regulate some signaling molecules and hormones. Thus, calorie restriction led to a statistically significant decrease in leptin and an increase in adiponectin levels, but the level of IGF-1 was not altered in former athletes [87]. Increased levels of adiponectin can be associated with anti-inflammatory effects due to the suppression of TNF and IFN-γ production [74]. It is assumed that the effects of calorie restriction on metabolism can be regulated by gene variants. The rs266729 variant of the adiponectin gene was associated with a significant increase in adiponectin levels and a decrease in the low-density lipoprotein, cholesterol and insulin levels, as well as a homeostasis model assessment for insulin resistance after implementing a calorie-restricted diet [88]. As was previously mentioned, leptin stimulates pro-inflammatory cytokine production, so a restricted leptin level is associated with the anti-inflammatory effects of calorie restriction.

Long-term calorie restriction reduced the serum thyroid hormone T3 levels. This effect is suspected to be connected with the diet-induced increase in life span, because a decreased T3 concentration leads to the conservation of energy and a lower free radical production [89]. The long-term effects of calorie restriction include a decrease in the serum total and free testosterone levels [90], which can play an important role in the initiation and development of obese-associated diseases, for example, cancer.

### 3.5. Cell Processes

Calorie restriction can take part in the regulation of various important cell processes. The stimulation of eIF2 signaling by the restriction of protein and calories led to the blockage of protein synthesis and cell growth. Moreover, the decrease in protein and calories in food inhibited the cell cycle at the G2M phase [84]. Calorie restriction also led to a lowering of sE-selectin [91]. sE-selectin is an adhesion molecule expressed on the endothelial surface in response to pro-inflammatory stimuli. It was revealed that sE-selectin promotes tumor progression through the activation of adhesion and migration [92]. It was also revealed that calorie restriction can influence fibrosis and endothelial function. Thus, a calorie-restricted diet led to an improvement in the biomarkers associated with fibrosis (MMP2, 8 and 9; CHI3L1; and PAI1), endothelial function (endostatin, ET1 and VEGFR1) and atrial stretch (NTpro-ANP) in patients with type 2 diabetes [79]. A very-low-calorie diet decreased the fibroblast growth factor 21 levels, which is supposed to be a predictor of treatment efficiency in patients with non-alcoholic fatty liver disease [93]. Calorie restriction can increase the levels of NAD^+^ levels. NAD^+^ is an essential substrate for the sirtuin family, which comprises well-known mediators of calorie restriction effects [94].

Thus, calorie restriction can initiate alterations to metabolism and take part in the regulation of various pathologic processes.

## 4. Calorie Restriction as a Treatment Strategy

Since calorie restriction can influence various physiological processes, there are many investigations proving that a calorie-restricted diet can improve the state of patients with various diseases.

### 4.1. Type II Diabetes and Cardiovascular Diseases

Calorie restriction in patients with type II diabetes resulted in improvements to their quality of life, weight and liver function, as well as a decrease in lipid, glucose and HbA_1c_ levels [95]. Calorie restriction improved their skeletal muscle insulin resistance (IR) and skeletal muscle and hepatic oral disposition index (DI), a measure of pancreatic insulin secretion. The improvement in β-cell function due to a calorie-restricted diet could be used for the prevention of type 2 diabetes. In obese patients with type 2 diabetes, calorie restriction decreased the glomerular filtration rate (GFR), which is associated with an improvement in glomerular hyperfiltration. The diet also reduced the levels of such cardiovascular risk factors as HbA1c, blood glucose and serum triglycerides, while the concentration of ApoA-I was increased [96]. A very-low-calorie diet led to a reduction in the cardiometabolic risk in women who were overweight/obese. This included a decrease in systolic blood pressure and a reduction in their total cholesterol, LDL cholesterol and triglycerides [97]. It is known that patients with cardiovascular diseases have an exceedingly active sympathetic nervous system. The key metabolic alterations associated with this peculiarity are obesity and insulin resistance [98]. An evaluation of the influence of calorie restriction on cardiovascular risk factors in healthy men and women showed that a low-calorie diet led to a decrease in noradrenaline levels; this effect was positively correlated with the serum total cholesterol, LDL-c and apolipoprotein B levels [99]. The predictors of cardiovascular disease include arterial stiffness. A low-calorie diet improved arterial stiffness, which was correlated with insulin sensitivity and such inflammatory markers as the C-reactive protein, interleukin 8 and tumor necrosis factor alpha [100].

### 4.2. Non-Alcoholic Fatty Liver

A very-low-calorie ketogenic diet in patients with non-alcoholic fatty liver disease (NAFLD) led to weight loss and significant reductions in visceral adipose tissue and the liver fat fraction [93]. A calorie-restricted and low-fat diet lowered the triglycerides, total and LDL cholesterol, liver enzymes, fasting glucose, insulin and HOMA-IR index in individuals with NAFLD, which reflects an improvement in obesity-related cardiometabolic alterations involved in the development of hepatic steatosis [101]. Liver biomarkers indicate hepatic health and are linked to cardiometabolic disease. Calorie restriction decreased the alanine aminotransferase and gamma-glutamyl transferase levels in male participants. This result suggests that CR can improve liver and metabolic disease risk factors [102].

### 4.3. Chronic Kidney Disease

Inflammation plays an important role in chronic kidney disease progression. Obesity promotes the production and activity of pro-inflammatory factors. Calorie restriction in patients with moderate to severe CKD had significant benefits to body weight and fat mass, which were followed by an improvement in metabolic health markers, including markers of oxidative stress and inflammatory responses [82]. A very-low-calorie diet improved the kidney function in patients with obesity and mild kidney failure. Moreover, a calorie-restricted diet significantly decreased the uric acid and ferritin levels. Ferritin is known as a marker of inflammation [103]. In patients with obesity and hyperuricemia, a calorie-restricted diet led to a significant decrease in their uric acid levels [104].

### 4.4. Metabolic Syndrome

Calorie restriction in older adults with metabolic syndrome led to a decrease in their triglyceride and apoC-III levels, which demonstrated an improvement in lipid metabolism [105]. It was revealed that metabolic syndrome and closely related vascular disorders such as diabetes and hyperlipidemia, as well as immune or mitochondrial dysfunction and oxidative stress, play an important role in the pathogenesis of multiple sclerosis (MS). Improvements in multiple sclerosis patient health after implementing a calorie-restricted diet were assumed to be connected to its anti-inflammatory and neuroprotective effects [106].

## 5. Calorie Restriction under Malignant Tumors

### 5.1. Effects of Calorie Restriction in Laboratory Models with Malignant Tumors

Recently, a lot of investigations have been devoted to the effects of calorie restriction on cancer development in vitro and in vivo. The combined action of calorie restriction and a glutamine antagonist, 6-diazo-5-oxo-L-norleucine (DON), in a glioblastoma mouse model influenced glutaminolysis and decreased the glucose levels, resulting in the initiation of cancer cell death and an improvement in overall mouse survival. It is important to stress, however, that the mice used for these studies were not treated with surgery, radiation or standard chemotherapy, so it is unclear if a similar CR effect would be revealed in patients with glioblastoma [107]. A ketogenic diet in combination with calorie restriction led to the suppression of neuroblastoma cell growth and a significant decrease in tumor blood vessel density, which were accompanied by the up-regulation of AMP-activated protein kinase and decreased serum levels of essential amino acids in CD-nu mice with neuroblastoma xenografts. The influence of calorie restriction on energy metabolism promoted the anti-tumor and anti-angiogenic effects of chemotherapy [108]. It is interesting that calorie restriction in AKR/J female mice with a relatively short lifespan, because of leukemia development around 6 months of age, led to an increased survival rate of offspring [109]. Calorie restriction decreased the levels of glucose, cholesterol, triglycerides, low-density lipoproteins and non-esterified fatty acids and prevented the development of radiation-induced intestinal cancer in C3B6F1 *Apc*^Min/+^ mice [110]. A mouse model of triple-negative breast cancer was used for an evaluation of the influence of obesity and calorie restriction on breast cancer development. Obese C57BL/6 mice were orthotopically injected with E0771 cells. Calorie restriction decreased the expression of genes associated with the epithelial-to-mesenchymal transition. Furthermore, the expression of several genes in the normal mammary tissue that regulate hypoxia and reactive oxygen species production, and p53 signaling was also altered. The tumor weight, systemic cytokines and the incidence of lung metastases were decreased in calorie-restricted mice [111]. Non-small-cell lung cancer cells with the KRAS mutation and LKB1 loss demonstrated alterations to the activity of the enzymes involved in the regulation of glycolysis, glutaminolysis, tryptophan catabolism and phospholipid metabolism. Calorie restriction resulted in a decrease in cancer cell proliferation and an enhanced sensitivity to chemotherapy [112]. Glucose restriction combined with a curcumin treatment led to the inhibition of Na(+)-H(+) exchanger-1 (NHE1), vATPase, monocarboxylate transporter (MCT)-1, the MCT4 level and proton-extruding enzymes in hepatoma cells, with an intracellular pH reduction. It was revealed that the production of ATP and lactate decreased according to the pH change. Furthermore, this combined treatment initiated structural changes in the proteins of the mTOR signaling pathway. The consequences of this influence were the inhibition of cancer cell migration and proliferation [113]. Three types of preclinical models of colorectal cancer have been used to investigate the effects of calorie restriction on cancer development: carcinogenic-induced models with chemicals, the models with transplantation of CRC cell lines and genetically modified mouse models of CRC. It was revealed that all models are useful tools to investigate cellular and molecular response to calorie restriction. Nevertheless, it is necessary to stress that there are nutritional recommendations for cancer care, and weight-loss or reduction of protein may enhance risk of malnutrition or sarcopenia in cancer patients [114].

### 5.2. Effects of Calorie Restriction in Patients with Malignant Tumors

The influence of a calorie-restricted diet on patients with various types of cancer was also investigated. Short-term fasting in glioma patients led to significant metabolic changes, including decreased glucose and insulin levels, which can be treated as candidate markers for a better prognosis [115]. Short-term calorie restriction during chemotherapy in patients with diffuse large B-cell lymphoma led to better results and initiated a significant decrease in cancer cell proliferation [116].

One of the causes of chemotherapy resistance is diminished blood circulation and the formation of a hypoxic cancer microenvironment. This process is connected to anaerobic energy metabolism. It was revealed that calorie restriction reduced glycolysis in metastatic breast cancer patients and improved the effectiveness of treatment [117]. Calorie restriction prevents cardiotoxicity after anthracycline chemotherapy in patients with malignant tumors. This effect is associated with the modulation of apoptosis, inflammation, oxidative stress and endothelium-dependent vasodilation by diet [118]. The effects of calorie restriction on cancer development and progression are based on alterations to metabolism associated with various signaling pathways that are up- or down-regulated. Calorie restriction reduced glycemia, IGF-1 and the IGF-1/IGFBP3 ratio, and improved insulin sensitivity in patients with Barrett’s esophagus (BE). These alterations were associated with the down-regulation of the insulin/IGF-1 and ERK signaling pathways, which, in turn, decreased the risk of esophageal adenocarcinoma development [119]. Protein and calorie restriction altered the pharmacokinetics of irinotecan and improved the therapeutic window in patients with liver metastases of solid tumors treated with irinotecan [120].

Calorie restriction can influence melanoma development. Le Noci et al. showed that the use of caloric-restriction mimetics in mouse models of melanoma B16 reduced metastasis in the lungs. This effect was connected to the modulation of the immune microenvironment, including a significant increase in alveolar macrophages and CD103+ dendritic cells; an increase in M1 and a decrease in M2 markers in myeloid cells; the activation of CD3 T lymphocytes and NK cells; and an increased cytotoxic activity of effector cells in the lung [121]. The combined action of ropivacaine-loaded liposomes and calorie restriction effectively repressed melanoma B16 development and relieved cancer pain. These effects were associated with autophagy suppression and a reduction in the VEGF-A levels [122].

Thus, both obesity and calorie restriction regulate various signaling pathways associated with cancerogenesis, which can promote or suppress malignant tumor development (Figure 2).

The underlying mechanisms of calorie restriction influence on cancerogenesis is still unclear and can be associated with not only metabolic alterations. The CR effects can be connected with the regulation of splicing factors that leads to TORC pathway suppression [123]. Fasting reduces monocyte mobilization, which decreases the activity of inflammation via AMPK activation and suppression of CCL2 production [124]. CR can modulate enzyme activity. For example, decreased stearoyl-CoA desaturase (SCD) leads to imbalance between unsaturated and saturated fatty acids that can slow tumor growth [125]. The peculiarities of metabolism connected with race and age of patients can determine the CR effectivity [126]. On the other hand, starvation can increase mTORC1 signaling activity via deregulation of MiT/TFE-RagD-mTORC1-MiT/TFE feedback circuit [127]. Functional caloric restriction in T-cells triggered by elevated potassium ion levels lead to T cell disfunction and stemness [128].

## 6. Conclusions

The strong connection between obesity and cancer development is based on various alterations including alterations of metabolism. Increased levels of glucose, fructose and lipids can be connected with not only increased food uptake but also with altered expression of factors regulating metabolic processes under obesity. Obesity is associated with abnormalities of IGF axis and increased estrogen levels. Low-grade chronic inflammation, deregulation of adipokines levels and hypoxia connected with obesity play a crucial role in cancer initiation and progression. Calorie restriction is one of the effective weight-loss strategies. A calorie-restricted diet improves lipid, carbohydrate and protein metabolism, and decreases oxidative stress and inflammation. It is shown that fasting can be successfully used in patients with various diseases, including type II diabetes, cardiovascular diseases, non-alcoholic fatty liver, chronic kidney disease and metabolic syndrome. There are many investigations devoted to the influence of calorie restriction on cancerogenesis. It was revealed that different calorie-restricted regimes can suppress the development of such types of malignant tumors as glioblastoma, leukemia, intestinal cancer, breast cancer, lung cancer, hepatoma and colorectal cancer in preclinical models. Calorie restriction can improve the effectiveness of chemotherapy, but the use of fasting in clinical practice has limitations. It is interesting to stress that obesity and calorie restriction can modulate the activity of the same signaling pathways with opposite results. Nevertheless, further investigations examining the mechanisms of calorie restriction are necessary for the creation of new, adjuvant methods of therapy.

## Figures and Tables

**Figure 1 ijms-24-09601-f001:**
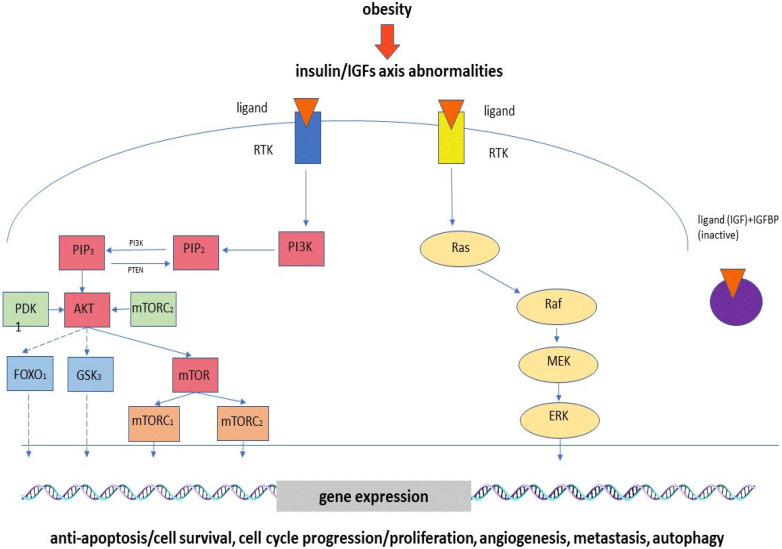
The influence of IGF axis abnormalities on cancer development.

**Figure 2 ijms-24-09601-f002:**
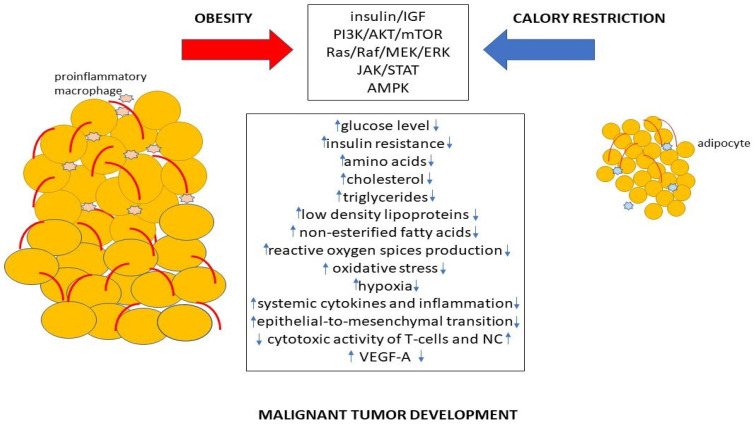
The mechanism of promotion or suppression of cancer development by obesity and calorie restriction.

**Table 1 ijms-24-09601-t001:** Signaling pathways involved in cancer initiation under obesity.

Types of Cancer	Signaling Pathway	Genes	Up-Regulation or Down-Regulation	Effects	References
Esophageal cancer	IGF/PI3K/AKT	IGFBPL1	Down-regulation	Inhibition of esophageal cancer cell clonal formation and proliferation; induction of cell apoptosis and G1/S phase arrest	Liu 2020 [28]
Hepatocellular carcinoma	IGF/PI3K/AKT;	IGF1	Up-regulation	Stimulation of DNA synthesis and cyclin D1 expression, inhibition of proteasome-mediated cathepsin B (CTSB) degradation in hepatocellular carcinoma cells;	Adamek 2018 [30]
		IGF2	Up-regulation	Stimulation of neoangiogenesis in hepatocellular carcinoma;	
Adrenocortical adenoma	IGF/JAK-STAT	IGF1	Up-regulation	Stimulation of adrenocortical adenoma cell growth, differentiation, proliferation and survival	Lazúrová 2020 [36]
		IGF2	Up-regulation	Stimulation of adrenocortical adenoma cell growth, differentiation, proliferation and survival	
Colorectal cancer	IGF-1/IGFBP	VEGF	Up-regulation	Stimulation of neoangiogenesis in colorectal cancer	Ciulei 2020 [35]
Colon cancer	IGF/PI3K/AKT/mTOR/MAPK	IGFs	Up-regulation	Stimulation of colon cancer cell growth, differentiation, proliferation and survival	Giovannucci 2001 [37]
Melanoma	IGF/AKT	IGFs	Up-regulation	Stimulation of autophagy in melanoma cells	Wang 2018 [40]
Breast cancer	IL-6/JAK/STAT3	IL-6	Up-regulation	Stimulation of breast cancer cell proliferation and invasiveness, suppressing apoptosis	Manore 2022 [60]
	HIF1α	HIF1α	Up-regulation	Stimulation of breast cancer cell proliferation	Brown 2021 [70]
	p53	p53	Down-regulation	Blockage of breast cancer cell apoptosis	Brown 2021 [70]
	AMPK	AMPK	Up-regulation	Stimulation of breast cancer cell proliferation	Wang 2017 [25]
Endometrium cancer	ER/IGF	IGF1	Up-regulation	Stimulation of endometrium cell proliferation and inhibition of apoptosis	Shaw 2016 [46]

## Data Availability

Not applicable.

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
