# Peer review of "Effects of Obesity and Calorie Restriction on Cancer Development"

_ijms, 2023, doi:10.3390/ijms24119601_

Round 1
Reviewer 1 Report
Ekaterina Sergeeva et al. described and summarized signaling pathways and key gene regulation by obesity and calorie restriction in vitro and in vivo cancer studies. Although it is an important and interesting topic, the structure and contents of the thesis are somewhat monotonous and broad. Figures and Tables are also broad and non-specific.
1. Authors can specify tissues and organs that regulate cancer cell phenotypic conversion in obese conditions or caloric restriction. This can be an important issue for crosstalk between tissues, organs, and cancer cells. Sometimes specific metabolites and hormones mediate these crosstalks.
2. Specific cancers, such as colorectal cancer or breast cancer, are tightly associated with obesity or caloric restriction. Authors can assign cancer types based on specific signaling pathways and key genes in obesity, calorie restriction, inflammation, and immune response.
3. There is a big difference between in vivo mouse studies and clinical human studies. It will be helpful for the reader if the authors explain the differences between mice and humans in at least some of the important studies.
4. Authors can focus on IGF signaling and redraw its cellular process instead of the broad and general current Figure 1/3.
5. The author needs to take a look at important recent papers on this topic. PMID number: 34671163, 27919065, 28619945, 30923193, 31442403, 32877581
Author Response
Dear reviewer, thank you for reviewing of our manuscript.

Reviewer 2 Report
The manuscript "Effects of Obesity and Caloric Restriction on Cancer Development" aims to determine the effect of obesity and caloric restriction in cancer, mainly the metabolic alterations the promote cancer development and the mechanisms associated with caloric restriction.
The manuscript relies os the metabolic alterations caused by obesity, being the obesity the trigger that develops cancer and the second part of the review approaches the influence of caloric restrition in cancer development. The structure of the manuscript is interesting (there are manuscripts published with the effect of calorie restriction in cancer prevention and therapy). The review is interesting and novel. The mechanisms associated with obesity are novel and well described. It would be interesing to have the clinical trials associated with some specific molecules that are described in the manuscript.
The content of the manuscript is relevant, but the figures are really poor and some signalling pathways or figures will improve the manuscript. A final image that demonstrates the dual effect of the reviwe will also benefit the manuscript.
The conclusions do not demonstrate the most relevant ideas of the manuscript. Please improve it.
Author Response

(The authors gave the same response as above.)

Round 2
Reviewer 1 Report
Overall, the authors have updated based on comments from reviewers. One thing to point out is that Figure 1 is confusing. Is the orange ligand insulin/IGF? RTK signaling can crosstalk JAK-STATA by SH2 domains. However, insulin/IGF is basically for RTKs, not GPCRs and cytokine receptors-mediated JAK/STAT. Current Figure 1 does not provide an in-depth look at IGF1 signaling nor good information about the link between obesity and cancer.
Author Response
Dear reviewer, thank you for your attention to our manuscript.
